 

# Control of lipid domain organization by a biomimetic contractile actomyosin cortex

**Sven Kenjiro Vogel[1]\*[†], Ferdinand Greiss[1,2,3†], Alena Khmelinskaia[1,3], Petra Schwille[1]\***

[1]Max-Planck Institute of Biochemistry, Martinsried, Germany; [2]Systems Biophysics, Physics Department, Ludwig-Maximilans-University, Munich, Germany; [3]Graduate School of Quantitative Biosciences, Ludwig-Maximilans-University, Munich, Germany

**Abstract** The cell membrane is a heterogeneously organized composite with lipid-protein micro-domains. The contractile actin cortex may govern the lateral organization of these domains in the cell membrane, yet the underlying mechanisms are not known. We recently reconstituted minimal actin cortices (MACs) (Vogel et al., 2013b) and here advanced our assay to investigate effects of rearranging actin filaments on the lateral membrane organization by introducing various phase-separated lipid mono- and bilayers to the MACs. The addition of actin filaments reorganized membrane domains. We found that the process reached a steady state where line tension and lateral crowding balanced. Moreover, the phase boundary allowed myosin driven actin filament rearrangements to actively move individual lipid domains, often accompanied by their shape change, fusion or splitting. Our findings illustrate how actin cortex remodeling in cells may control dynamic rearrangements of lipids and other molecules inside domains without directly binding to actin filaments.

**\*For correspondence:** svogel@ biochem.mpg.de (SKV); schwille@ biochem.mpg.de (PS)

[†]These authors contributed equally to this work

**Competing interests:** The authors declare that no competing interests exist.

## Introduction

The spatiotemporal organization of lipids, proteins and other molecules at and within the cell membrane is pivotal for many fundamental cellular processes, such as signal transduction from the extracellular to the intracellular space (*Groves and Kuriyan, 2010*). Recent findings suggest that the cell membrane should be considered as a heterogeneous lipid protein layer with coexisting small micro domains and clusters of lipids and proteins that are assumed to dynamically form and reorganize in response to external and internal cues (*Engelman, 2005*; *Simons and Gerl, 2010*). Whether and how their spatiotemporal organization is actively regulated and maintained by the cell remains to be revealed. In vivo and in vitro studies suggest an important role of the eukaryotic actin cytoskeleton that interacts with the cell membrane via membrane-associated proteins (*Heinemann et al., 2013*; *Köster et al., 2016*; *Murase et al., 2004*; *Sheetz et al., 1980*). Actin structures were found to mediate the lateral organization of membrane proteins (*Gudheti et al., 2013*) and to modulate their diffusive behavior (*Heinemann et al., 2013*; *Honigmann et al., 2014*; *Murase et al., 2004*). Theoretical considerations have proposed a key role of the actin motor myosin for organizing and forming distinct protein and lipid micro-domains in cell membranes (*Gowrishankar et al., 2012*). However, direct experimental evidence for a control of lipid micro-domains by actomyosin rearrangements is still lacking.

In eukaryotic cells, the actin cortex is constantly rearranged by the motor protein myosin II and dozens of actin-binding proteins. Therefore, reducing the complexity of experimental conditions, e.g. reducing dimensionality or exploiting a minimal biomimetic system, and utilizing microscopic techniques with a high temporal resolution is beneficial for studying these processes. Phase-separated lipid bilayers with controlled lipid compositions are well-established test beds to mimic lipid

micro-domains in cell membranes and biological processes, e.g. the lateral organization of proteins that are otherwise difficult to observe in vivo. Ternary mixtures of lipids below their characteristic melting temperature (Tm) can phase separate into liquid disordered ($L_d$) and liquid ordered ($L_o$) domains. Liu and Fletcher used phase-separated giant unilamellar vesicles (GUVs) as a model system to study the influence of branched actin networks on the miscibility transition temperature (*Liu and Fletcher, 2006*). They reported that the formation of localized actin networks on PIP2-containing phase-separated GUVs could act as switch for the orientation of lipid domain growth sites.

However, the effects of an actin meshwork on individual lipid domains are technically difficult to study, due to the three-dimensional architecture of the GUVs. We therefore made use of a minimal biomimetic system of planar geometry that we recently developed (*Vogel et al., 2013a*, *2013b*) and combined the minimal actin cortex (MAC) with supported phase-separated membranes and free-standing lipid monolayers. We then visualized and studied the effects of actin filament adhesion and myosin-driven rearrangements on the lateral organization of membrane domains with total internal reflection microscopy (TIRFM) and confocal spinning disk microscopy.

## Results

### Actin filament partitioning on phase-separated membranes

The effects of adhesion and myosin-induced contraction of an actin meshwork on the lateral organization of lipid domains was studied by combining phase-separated lipid membranes with our established assay featuring contractile MACs (*Vogel, 2016*; *Vogel et al., 2013a*, *2013b*) (*Figure 1A*). For the adhesion assay, a ternary lipid mixture with DOPC, DPPC and Cholesterol in a 1:2:1 molar ratio was prepared (see Material and methods). Similar lipid compositions were described to form $L_o$ and $L_d$ phases in free-standing membranes up to 30°C (*Veatch and Keller, 2003*, *2005*). The low miscibility transition temperature of the mixture avoids thermal degradation of proteins and allowed us to study the phase transition behavior in the presence of actin filaments. The fluorescent probe DiD (0.03 mol%) was used to label the $L_d$ phases (*García-Sáez et al., 2007*). The density of biotinylated actin filaments was controlled by the concentration of biotinylated lipid DSPE-PEG(2000)-Biotin (see also [*Vogel, 2016*; *Vogel et al., 2013b*]). As observed with TIRF microscopy, the membrane separated into micrometer-sized $L_o$ and $L_d$ domains (*Figure 1B–E*) and homogenized at ~37°C. The observed shift of 7°C in Tm compared with studies from *Veatch and Keller (2003)* agreed with our expectations considering the interaction of lipid molecules with the mica support (*García-Sáez et al., 2007*).

The adhesion of actin filaments via neutravidin to biotinylated DSPE provided a stable link between the MAC and the lipid bilayer over a wide range of temperatures. To validate the partitioning preference of all molecular components, the fluorescence signals of labeled neutravidin and actin filaments were acquired after domain formation when cooled below Tm (*Figure 1B–E*). We found that both neutravidin and actin filaments partitioned into the $L_o$ phase (*Figure 1B and C*). We therefore concluded that the biotinylated lipid DSPE partitioned strongly to the $L_o$ phase.

### Actin filament crowding effects on supported phase-separated membranes

We prepared low (0.01 mol % DSPE-PEG(2000)-Biotin) and medium (0.1 mol % DSPE-PEG(2000)-Biotin, *Figure 2B*) dense MACs to investigate the effect of actin filament density on phase-separated membranes. To this end, the ternary lipid mixture was incubated with non-labeled neutravidin and placed on a temperature-controlled microscope objective. A fluorescence image was acquired as a reference before actin filaments were added (*Figure 2A and B* (left column)). A homogeneous membrane was produced by heating the microscope objective to 42°C (above Tm). Actin filaments were added to the membrane above Tm (*Figure 2A and B* (middle column)), and the sample was subsequently incubated for ~45 min at 42°C (*Figure 2A*). The membrane was completely covered by actin filaments after approximately 30 min (*Figure 2A and B* (middle column)). As final step, the membrane was slowly cooled down to 30°C without active cooling and the fluorescence signal was recorded after complete phase separation (*Figure 2A and B* (right column)). While the low-density MAC did not show any influence on membrane domain properties (data not shown), the medium-dense MAC caused the formation of smaller domains (*Figure 2B*, right column). Both, actin filaments

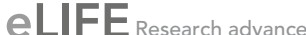

**Figure 1.** Minimal actin cortex on supported phase-separated lipid membranes. (**A**) Illustration depicting the preparation of the SLB and the coupling of actin filaments to the membrane (adapted from [*Vogel et al., 2013a*]). (**B**) Since DiD is known to partition specifically to the $L_d$ phase (1:2:1 DOPC: DPPC:Cholesterol), the fluorescence signal of Alexa-488-phalloidin-labeled actin filaments showed that DSPE-PEG(2000)-Biotin partitioned into the $L_o$

*Figure 1 continued on next page*

*Figure 1 continued*

phase. (**C**) Likewise, Oregon-Green-labeled neutravidin colocalized with the $L_o$ phase. (**D and E**) Line profiles of the normalized fluorescence signal of the actin cortex, DiD and neutravidin as was measured along the superimposed arrows in (**B**) and (**C**). Scale bar, 10 μm.

and DSPE-PEG(2000)-Biotin partitioned into the $L_o$ phase (*Figure 2B and C*), confirming our initial observations (*Figure 1B–E*). We conclude that actin filaments act as nucleation sites for domain formation and drive their lateral spatial distribution reproducing the observations reported by *Honigmann et al. (2014)*.

To further investigate the relation between lateral membrane organization and the binding of actin filaments, we exposed high-density MACs containing a lipid mixture with 1.0 mol% biotinylated DSPE to different temperatures below Tm (*Figure 2C–F*). Fluorescently labeled actin filaments were added to the sample chamber at specified temperatures below Tm and imaged for ~30 min using TIRF microscopy (*Figure 2C*). Because of the strong binding affinity of neutravidin and biotin, we found that the change in fluorescence intensity of labeled actin filaments (which we used as a measure for actin filament binding) was independent of the applied temperature (*Figure 2F*). Hence, the adhesion process of actin filaments to the membrane was constant over the employed temperature range. The contour length (L) (*Figure 2G*) of membrane domains was extracted and tracked over time (*Figure 2D*). After ~1 min of actin filament addition, first domain deformations could be observed and the contour length was found to grow until a steady state at ~20 min was reached (*Figure 2D*). The final contour length increased with temperature (*Figure 2E*, final change in contour length: $L_{Final}$ = 188 ± 4 μm and τ = 9.5 ± 0.5 min at 24°C, $L_{Final}$ = 314.0 ± 1.5 μm and τ = 5.6 ± 0.1 min at 29°C, $L_{Final}$ = 706.3 ± 2.5 μm and τ = 4.4 ± 0.1 min at 32°C). The line tension γ between $L_o$ and $L_d$ domains is known to be a linear function of temperature with $\gamma \approx \gamma_0(T_C - T)/T_C$, where $T_C$ is the critical temperature (*Baumgart et al., 2003*; *Honerkamp-Smith et al., 2008*; *Veatch et al., 2008*). With the temperature-independent actin filament binding and the boundary energy given by $E = \gamma L$ (*Yang et al., 2016*), where $E$ is the boundary energy, the final contour length $L$ should increase with

$$L = \frac{E}{\gamma_0} \frac{T_C}{T_C - T}. \tag{1}$$

The model describes the experimental data well (*Figure 2E*) and gives a length change ($E/\gamma_0$) of 86 ± 9 μm at $T$ = 0°C with $T_C$ = 37°C.

## Actomyosin contraction governs lateral membrane domain organization

Aster shaped actomyosin clusters form in vivo (*Luo et al., 2013*; *Munro et al., 2004*; *Verkhovsky et al., 1997*) and in vitro (*Backouche et al., 2006*; *Köster et al., 2016*; *Murrell and Gardel, 2012*; *Soares e Silva et al., 2011*; *Vogel et al., 2013b*) upon myosin's contractile activity. Synthetic myofilaments contract the MAC in the presence of ATP and, hence rearrange actin filaments to form stable actomyosin clusters (*Figure 3A,C and G*; *Videos 1* and 5). For studying the effects of myofilament-induced actin rearrangements, we prepared two lipid mixtures for lipid mono- and bilayer experiments, namely DOPC, DPPC and Cholesterol in a 1:2:1 molar ratio and DOPC, PSM and Cholesterol in a 3:3:1 molar ratio containing 0.1 or 1 mol% DSPE-PEG(2000)-Biotin DSPE (see Material and methods).

We found that upon contraction, membrane domains deformed at the phase boundaries within minutes and resulted in various splitting and fusion events (*Figure 3A,D,F and G*; *Videos 2–4* and *6*). Domain shape changes, such as inward ingression, were observed and correlated with the movement of actomyosin clusters (*Figure 3D*; *Videos 3* and *4*). Actomyosin contraction against smaller $L_d$ domains resulted in their net movement and occasionally led to splitting events of the domain (*Figure 3D*; *Video 4*), while the contraction for $L_o$ domains led to deformation and fusion with neighboring domains (*Figure 3F*; *Video 6*). For both mixtures, the area of the $L_o$ phase remained constant over the course of actomyosin contraction (*Figure 3B and E*, left panels). The number of $L_d$ domains increased with time due to their splitting for the DOPC, DPPC, Cholesterol mixture (*Figure 3B*, right panel; *Videos 2–4*). The number of $L_o$ domains decreased in the DOPC, PSM, Cholesterol mixture due to their fusion and occasional disappearance during actomyosin contraction (*Figure 3E* (right

**Figure 2.** Impact of actin filaments on the lateral organization of phase-separated membranes. (**A**) Scheme of a phase separation experiment with and without MAC is shown with circles indicating time points when fluorescent images were acquired. (**B**) Size distribution of lipid domains changed in the presence of a MAC while undergoing phase separation. TIRF images of a medium-dense MAC with DiD-labeled $L_d$ phase (1:2:1 DOPC:DPPC: Cholesterol) are shown below Tm (30°C) (left column), above Tm (42°C) in the presence of bound Alexa-488-phalloidin-labeled actin filaments (middle

*Figure 2 continued on next page*

*Figure 2 continued*

column) and below Tm (30°C) in the presence of bound actin (right column). The final size of $L_d$ domains decreased through the presence of actin filaments. (**C**) Change in contour length through the binding of actin filaments to the already phase-separated membrane at various temperatures below Tm. A high-density MAC with a DiD-labeled $L_d$ phase before (0 min) and after the addition (32 min) of actin filaments at 24°C is shown. (**D**) Contour length of $L_d$ domains upon F-actin adhesion with time and mono-exponential decay fits at 24°C (blue dots), 29°C (green dots) and 32°C (red dots). (Note that the observed local dip in contour length between 2 and 5 min at 24°C is because of a focal misalignment and blurring of the images after the addition of actin filaments). (**E**) Final change in contour length for the different temperatures. With a constant lateral pressure, the linearly decreasing line tension with temperature leads to an increase in final contour length according to *Equation (1)*. Dashed vertical line indicates Tm of 37°C. (**F**) The binding kinetics of actin filaments was tracked with the normalized fluorescence signal over time at 24°C (blue dots), 29°C (green dots) and 32°C (red dots). (**G**) Illustrative sketch for the change in contour length before and after F-actin adhesion. Scale bars, 10 μm.

panel) and G; *Videos 6* and *7*). A similar behavior in $L_o$ domain disappearance has been observed by *Köster et al. (2016)*. In contrast to their work, we did not observe a decrease of the total $L_o$ phase domain area in neither of the lipid mixtures (*Figure 3B and E*, left panels). In summary, we showed active lateral reorganization of lipid domains for two different lipid mixtures, indicating the general validity of our experimental observations.

In order to mimic free-standing membranes without support-induced friction as in the SLBs but keeping the technical advantages of a planar geometry, we made use of a recently developed lipid monolayer system (*Chwastek and Schwille, 2013*) with an air-liquid interface (*Figure 4A–D*). Actin filaments were coupled to ternary phase-separated lipid monolayers similar to the situation in the supported lipid bilayer (SLB) system via biotinylated lipids (1 mol%) and the use of neutravidin. Contrary to the situation in the SLB system, actin filaments preferentially bind to the liquid extended (disordered) ($L_e$) phase where also the neutravidin anchor protein mainly partitioned to (*Figure 4A–D*). We observed this behavior for both lipid mixtures that were also used for the SLBs (see Material and methods). In low- and medium-dense monolayer MACs the liquid condensed (ordered) $L_c$ domains acquire circular shapes and the actin filaments close to the phase boundaries align to their circular shape (*Figure 4A*). In the case of low and medium densities, we expect that the line tension energy dominates over the actin filament wetting energy at the phase boundary and hence the $L_c$ can assume an unrestricted shape with aligned actin filaments. Similar effects have been observed using bacterial cytoskeleton proteins (*Arumugam et al., 2015*). The addition of myosin filaments in the presence of ATP led to the contraction of the actin layer deforming and rearranging the $L_c$ domains in both lipid mixtures similar to what we observed in the SLB systems (*Figure 4B–D*; *Video 8*). The shape changes included fusion and stretching of the $L_c$ domains (*Figure 4B–D*; *Video 8*). Note that these 'active' deformation forces were only exerted in the presence of myosin filaments and ATP. Interestingly, the obtained lipid domain shapes seemed to be stabilized after the active contraction period, probably by the remaining actin filaments. Using the monolayer system also tells us that the frictional force caused by a solid support in SLBs does not play a significant role in the observed phenomena.

## Discussion

As cells may need to quickly adapt the macro-and microscopic organization of lipid and protein aggregates within the cell membrane due to external or internal cues, we propose that actomyosin-driven reorganization of actin filaments may aid to quickly govern their lateral distribution. Recent evidence exists that the presence of an actin meshwork influences the lateral diffusion behavior of lipids and proteins in membranes in vivo (*Murase et al., 2004*) and in vitro (*Heinemann et al., 2013*) and that it impacts the behavior of phase-separated membranes (*Honigmann et al., 2014*; *Liu and Fletcher, 2006*). The important role of myosin is further supported by recent theoretical and in vitro studies (*Gowrishankar et al., 2012*; *Köster et al., 2016*). It is therefore tempting to speculate that cortical actomyosin contractility may be a generic model for eukaryotic organisms not only to control their mechanical stability and shape but also to quickly and actively control the lateral lipid and protein organization at the cell membrane.

In our MACs that were combined with ternary lipid mixtures, we found that binding of actin filaments to a homogeneous bilayer at temperatures above Tm-induced spatial alignment of $L_o$ domains to actin-bound locations upon cooling below Tm (*Figure 2*). Here, actin filaments serve as

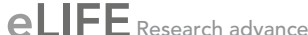

**Figure 3.** Actomyosin contraction governs lateral membrane organization for various lipid compositions. (**A**) TIRFM time-lapse images of a contracting MAC with Alexa-488-phalloidin labeled actin filaments and DiD-labeled $L_d$ domains of a 1:2:1 DOPC:DPPC:Cholesterol mixture. Scale bar, 10 μm. (**B**) For the 1:2:1 DOPC:DPPC:Cholesterol mixture the area of the $L_o$ phase was found to remain constant while the number of $L_d$ domains increased with time through splitting. (**C**) Distance to neighboring actomyosin clusters for the lipid composition of 1:2:1 DOPC:DPPC:Cholesterol. (**D**) Representative

*Figure 3 continued on next page*

*Figure 3 continued*

example (1:2:1 DOPC:DPPC:Cholesterol mixture) showing movement (white asterisks) and deformation (inward ingression, yellow arrows) of a phase boundary upon actomyosin contraction and its splitting into two separated domains (yellow arrowheads). Scale bar, 10 μm. (**E**) For a lipid composition of 3:3:1 DOPC:PSM:Cholesterol, the area of the $L_o$ phase remained again constant and the number of $L_o$ domains decreased. (**F**) Pulling of a $L_o$ domain (white asterisk) for a lipid composition of 3:3:1 DOPC:PSM:Cholesterol by actomyosin led to fusion (yellow arrowheads) and relaxation into a larger neighboring domain. Scale bar, 2.5 μm. (**G**) Time-lapse montage of the actomyosin contraction on a 3:3:1 DOPC:PSM:Cholesterol phase-separated membrane. Small $L_o$ domains are moved and fused to produce larger domains. Scale bar, 5 μm.

nucleation sites for domain formation and drive their lateral spatial distribution. Already phase separated, we could show that membrane domains reorganized with the adhesion of actin filaments depending on line tension and actin filament density. We further give direct evidence that the dynamic reorganization of actin filaments by myosin motors actively changes the macroscopic organization of membrane domains in our reconstituted phase-separated lipid bilayers and monolayers. We propose that the transition energy between the $L_d$ and $L_o$ phase (*Baumgart et al., 2003*; *García-Sáez et al., 2007*; *Honerkamp-Smith et al., 2008*) enabled the lipid anchored actin filaments to exert lateral forces on the phase boundaries leading to a macroscopic motion of lipid domains that eventually resulted in splitting, fusion or deformation of the lipid domains and in their overall change in number during and after the actomyosin contraction (*Figure 3B and E*, right panels; *Videos 2–4*, *6* and *8*). In a model, we would first consider a biotinylated lipid that is dragged by an actomyosin filament (*Figure 4E*). The actomyosin filaments are unequally associated with the lipid phases ($L_o$ and $L_d$) with different viscosities. The drag is counteracted by friction and, through the different viscosities, leads to domain deformation and rearrangement. Here, the force propagation would be independent of phase boundaries. As a second consideration, the

**Video 1.** Myosin-induced actin rearrangements in a minimal actin cortex (MAC) combined with a supported phase-separated lipid bilayer. MAC with a supported phase-separated membrane (1:2:1 DOPC:DPPC: Cholesterol) containing Alexa-488-phalloidin-labeled actin filaments (green) exhibits dynamic rearrangements of actin filaments after the addition of myofilaments in the presence of ATP and eventually forms actomyosin clusters. The phase-separated membrane containing DiD-labeled $L_d$ domains (red) is shown in the upper channel. The middle channel shows Alexa-488-phalloidin-labeled actin filaments (green) that bind to the $L_o$ domains. The lower channel shows the merge of both channels. TIRFM image sequence was acquired at 5 s. time intervals and contains 200 frames. The video is displayed at 15 frames per second (fps). Total time: 16.6 min. Scale bar, 10 μm. (Compressed JPG avi; 10.2 MB).

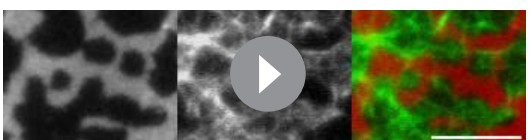

**Video 2.** Shape changes, rearrangements and fusion events of $L_d$ domains during actomyosin contraction. The phase-separated membrane (1:2:1 DOPC:DPPC: Cholesterol) containing DiD labeled $L_d$ domains (red) is shown in the left channel. The middle channel shows Alexa-488-phalloidin labeled actin filaments that bind to the $L_o$ domains. The right channel shows the merge of both channels. TIRFM image sequence was acquired at 5 s. time intervals and contains 124 frames. The video is displayed at 15 frames per second (fps). Total time: 10.3 min. Scale bar, 10 μm. (Compressed JPG avi; 0.8 MB).

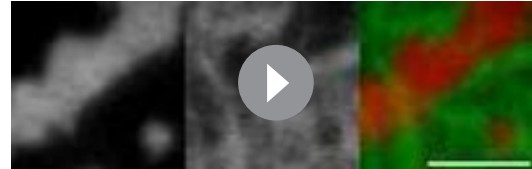

**Video 3.** Splitting, shape changes and deformations of $L_d$ domains during actomyosin contraction. The phase-separated membrane (1:2:1 DOPC:DPPC:Cholesterol) containing DiD labeled $L_d$ domains (red) is shown in the left channel. The middle channel shows Alexa-488-phalloidin-labeled actin filaments that bind to the $L_o$ domains. The right channel shows the merge of both channels. TIRFM image sequence was acquired at 5 s. time intervals and contains 200 frames. The video is displayed at 15 frames per second (fps). Total time: 16.6 min. Scale bar, 10 μm. (Compressed JPG avi; 0.8 MB).

**Video 4.** $L_d$ domain movement, splitting and ingression during actomyosin contraction. The phase-separated membrane (1:2:1 DOPC:DPPC:Cholesterol) containing DiD-labeled $L_d$ domains (red) is shown in the left channel. The middle channel shows Alexa-488-phalloidin-labeled actin filaments that bind to the $L_o$ domains. The right channel shows the merge of both channels. TIRFM image sequence was acquired at 5 s. time intervals and contains 200 frames. The video is displayed at 15 frames per second (fps). Total time: 16.6 min. Scale bar, 5 μm. Corresponds to *Figure 3D*. (Compressed JPG avi; 0.4 MB).

biotinylated lipid would need to overcome the transition barrier between phases while being dragged over a boundary (*Figure 4E*). Since our contraction experiments primarily showed that the boundaries deformed locally at actin filament sites and that actomyosin foci formed at the vicinity of phase boundaries, the high transition energy is favored as the dictating driving force for phase deformation. Furthermore, domain deformation was commonly found to happen when actin filaments were dragged over the $L_d$ phase that impedes the free contraction process through the strong partitioning preference of DSPE-PEG(2000)-Biotin to the $L_o$ phase.

With an estimated line tension of 1 pN for a 1:2:1 DOPC:DPPC:Cholesterol membrane (*Baumgart et al., 2003*; *García-Sáez et al., 2007*; *Kuzmin et al., 2005*; *Veatch and Keller, 2005*) and a reported value of ~1.5 $k_BT$ for the transfer of DPPC between phases (*Veatch et al., 2004*), the energy for one DPPC molecule to transition is roughly sixfold higher than the energy that is needed to elongate the phase boundary by 1 nm. Further experiments should elaborate on the physical model since actomyosin contraction is clearly a non-equilibrium process and line tension itself was reported to depend on the applied lateral tension (*Akimov et al., 2007*). Together with our experimental findings, we can conclude that actively rearranging actin filaments will lead to an extensive deformation of membrane domains through the physical link with a small subset of membrane constituents. A plausible mechanism is therefore apparent that explains active lateral rearrangements of membrane components by actomyosin contractions without the need of binding directly to actin filaments.

## Material and methods

### Actin labeling and polymerization

F-actin preparation was performed as described in *Vogel et al. (2013b)*. Briefly, a 39.6 μM actin solution (Actin/Actin-Biotin ratio of 5:1) was prepared by mixing rabbit skeletal actin monomers (32 μl, 2 mg/ml, Molecular Probes) with biotinylated rabbit actin monomers (1.6 μl, 10 mg/ml, tebu-bio/Cytoskeleton Inc.). F-buffer (1 mM DTT, 1 mM ATP, 10 mM Tris-HCl (pH 7.4), 2 mM $MgCl_2$ and 50 mM KCl) was added to the mixture in order to start polymerization. Actin polymers were labeled and stabilized with Alexa Fluor 488 Phalloidin according to the manufacturer's protocol (Molecular Probes). Finally, the 2 μM Alexa-488-Phalloidin-labeled biotinylated actin filament solution was stored at 4°C.

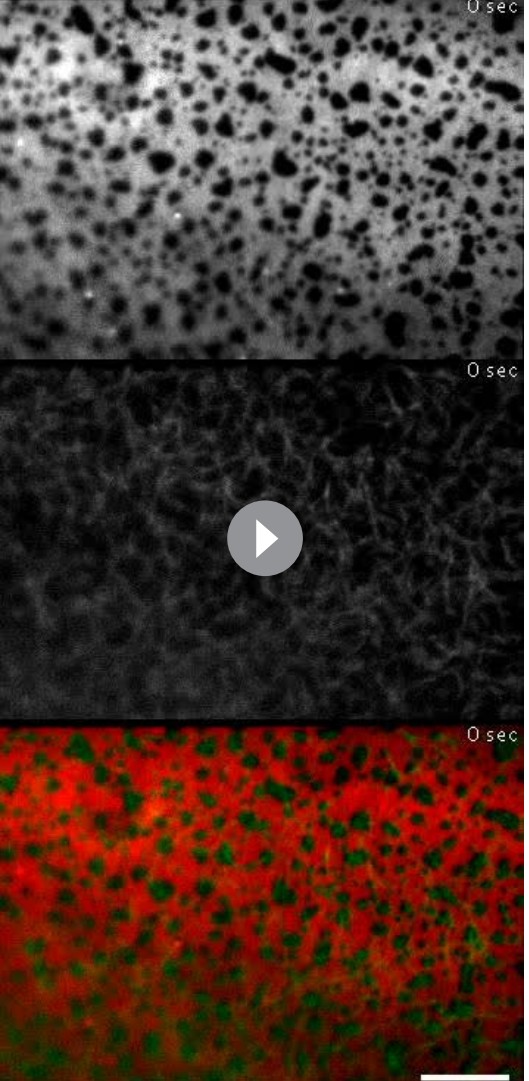

**Video 5.** Shape changes, rearrangements and fusion events of $L_o$ domains during actomyosin contraction. The phase-separated and DiD labeled (red) membrane (3:3:1 DOPC:PSM:Cholesterol) containing $L_o$ domains (dark) is shown in the upper channel. The middle channel shows Alexa-488-phalloidin-labeled actin filaments (green) that bind to the $L_o$ domains. The lower channel shows the merge of both channels. TIRFM image sequence was acquired at 5 s. time intervals and contains 500 frames. The video is displayed at 15 frames per second (fps). Total time: 41.6 min. Scale bar, 10 μm. (Compressed JPG avi; 28 MB).

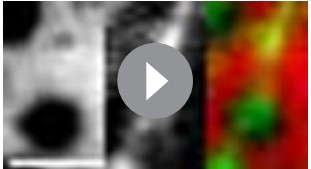

**Video 6.** Fusion event of a $L_o$ domain during actomyosin contraction. A small $L_o$ domain is stretched by actin filaments thereby pulling the domain toward a larger domain leading eventually to fusion of both $L_o$ domains. The phase-separated and DiD-labeled (red) membrane (3:3:1 DOPC:PSM:Cholesterol) containing $L_o$ domains (dark) is shown in the left channel. The middle channel shows Alexa-488-phalloidin-labeled actin filaments (green) that bind to the $L_o$ domains. The right channel shows the merge of both channels. TIRFM image sequence was acquired at 5 s. time intervals and contains 337 frames. The video is displayed at 120 frames per second (fps). Total time: 28 min. Scale bar, 2.5 μm. Corresponds to *Figure 3F*. (Compressed JPG avi; 0.4 MB).

## MAC (minimal actin cortex) preparation

1,2-Dioleoyl-*sn*-glycero-3-phosphorcholine (DOPC), 1,2-dipalmitoyl-*sn*-glycero-3-phospho-choline (DPPC), and cholesterol were added in a molar ratio of 1:2:1 or DOPC, N-palmitoyl-D-sphingomyelin (PSM), and Cholesterol in a molar ratio of 3:3:1 to a final lipid concentration of 5 mg/ml (Avanti Polar Lipids, Inc). The lipid bilayer was further supplemented with 0.03 mol% DiD (Molecular Probes, Eugene, OR) and 0.01, 0.1 or 1 mol% DSPE-PEG(2000)-Biotin (Avanti Polar Lipids, Inc.). The solution was dried under continuous nitrogen flux and then placed under vacuum for 1 hr to remove chloroform residuals. The pellet was rehydrated in SLB buffer (150 mM KCl, 25 mM Tris-HCl, pH 7.5) by vigorous vortexing and sonication.

20 μl of clear lipid suspension, was then diluted in 130 μl A-buffer (50 mM KCl, 2 mM $MgCl_2$, 1 mM DTT and 10 mM Tris-HCl, pH 7.5) and heated to 55°C. Meanwhile, freshly cleaved mica was fixated with immersion oil (Carl Zeiss, Jena, Germany) on a cover slip (22 × 22 mm, #1.5, Menzel Gläser, Thermo Fisher) and covered with the center part of a cut 1.5 mL Eppendorf tube. The Eppendorf tube was glued with UV-sensitive glue (Norland Optical Adhesive 63, Cranbury, USA). The chamber was filled with 75 μl of small unilamellar vesicles and incubated at 55°C for 45 min with 1 mM $CaCl_2$. Non-fused vesicles were removed by washing the suspension with 2 ml warmed A-buffer and gentle pipetting. The chamber's temperature was slowly cooled. Next, 2 μl of unlabeled or Oregon-Green-labeled neutravidin (1 mg/ml, Molecular Probes) diluted in 200 μl

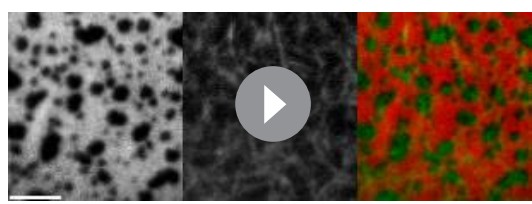

**Video 7.** Disappearance of small $L_o$ domains during actomyosin contraction. Small $L_o$ domains often vanish during actomyosin contractions. The phase-separated and DiD-labeled (red) membrane (3:3:1 DOPC:PSM: Cholesterol) containing $L_o$ domains (dark) is shown in the left channel. The middle channel shows Alexa-488-phalloidin-labeled actin filaments (green) that bind to the $L_o$ domains. The right channel shows the merge of both channels. TIRFM image sequence was acquired at 5 s. time intervals and contains 493 frames. The video is displayed at 120 frames per second (fps). Total time: 41 min. Scale bar, 5 µm. Corresponds to *Figure 3G*. (Compressed JPG avi; 4.1 MB).

A-buffer were added to the sample and incubated for 10 min. Finally, unbound proteins were removed by gently washing the solution with 2 ml A-buffer.

## Phase separation with MAC

The phase-separated membrane was heated with an objective heater (Carl Zeiss, Jena, Germany) to the setpoint of 42°C and cooled down to 30°C, both with and without Alexa-488-Phalloidin-labeled biotinylated actin filaments. The area distribution of domains at equilibrium (45 min after 30°C was reached) was then compared between the different concentrations of DSPE-PEG(2000)-Biotin (0.1, 0.01 mol%).

## Crowding-effect

The sample was placed on the TIRF microscope objective with attached objective heater and slowly warmed to the setpoint of 42°C. The lipid bilayer with 1.0% DSPE-PEG(2000)-Biotin was equilibrated at different temperatures below the melting point (~37°C for 1:2:1 DOPC:DPPC:Cholesterol). Of Alexa-488-Phalloidin-labeled biotinylated actin filaments, 20 µl were then carefully added to the chamber. Binding of actin filaments to the membrane was recorded by acquiring images in interleaved mode (488 nm and 640 nm) every 2.5 s, generating a time-lapse movie with a 5 s delay between subsequent images.

## F-actin network contraction by myofilaments

After the addition of 20 µl Alexa-488-Phallodin-labeled biotinylated actin filaments to the supported lipid bilayer at room temperature (~24°C), the mixture was incubated for approximately 45 min in order to ensure full binding of actin filaments to the membrane. Subsequently, residual actin was removed by gently exchanging the solution with 2 ml A-buffer. Once the properly assembled MAC was verified with TIRF microscopy, a solution of 20 µl myofilaments and 1 µl ATP (0.1 M) were added to start the compaction of actin filaments.

Images were acquired every 2.5 s in interleaved mode, which eliminated the cross talk between color channels of actin filaments and the phase-separated membrane.

## TIRF microscopy

Fluorescent imaging of labeled proteins and membrane was carried out on a custom-built TIRF microscope. The setup was integrated into an Axiovert 200 microscope (Zeiss). The probe was illuminated and imaged through a Plan-Apochromat 100x/NA 1.46 oil immersion objective with a 488 nm and 630 nm laser. Images were acquired with an Andor Solis EMCCD camera (electron gain = 300, exposure time = 50 ms, frame interval = 2.5 or 5 s) in interleaved mode.

## Data analysis

Image processing, analysis and data visualization were performed with Fiji and the scientific packages for Python. Multichannel beads were used to align double-color image stacks with the Fiji plugin Descriptor-based series registration. The contour length between the lipid domains was extracted by detecting edges (Canny edge detection; [*van der Walt et al., 2014*]) in single fluorescent images acquired from phase-separated membranes. The contour length over time was fitted to $L = L_{Final} \left(1 - \exp(-1/\tau \, (t - t_0))\right)$ with $t_0$ = 1 min. The actomyosin clusters were detected using the Laplacian of Gaussian method (scikit-image [*van der Walt et al., 2014*]) and assigned to neighbors using the Delaunay triangulation and its method vertex_neighbor_vertices.

To detect the lipid domains, the fluorescence images were firstly corrected for an uneven background signal. Then, the DiD signal was classified with a local adaptive thresholding algorithm and

**Figure 4.** Lipid monolayer domain shape changes and fusion events induced by actomyosin contractions. (**A**) Confocal spinning disk microscope images of Alexa-488-phalloidin-labeled actin filaments coupled to a ATTO655-DOPE-labeled lipid monolayer (3:3:1 DOPC:PSM:Cholesterol) in the absence of myofilaments. Actin filaments close to the phase boundaries of lipid monolayer domains align to their circular shape. (**B–D**) Spinning Disk microscope time-lapse images of Alexa-488-phalloidin-labeled actin filaments coupled to ATTO655-DOPE-labeled lipid monolayers during myofilament

*Figure 4 continued on next page*

*Figure 4 continued*

induced contractions of actin filaments. Actomyosin contractions lead to shape changes and fusion events (yellow arrows and arrowheads) of the lipid monolayer $L_c$ domains. (E) Scheme of a microscopic model. Scale bars, 10 µm.

the corresponding phase boundaries with the Canny edge detection algorithm. The binary image stacks were combined with the XOR operator and further refined using the morphological closing operation. Small holes were removed from the binary images. Small objects were detected with the Laplacian of Gaussian method and as a final step, combined with the binary images. The script was implemented in Python with the image processing package scikit-image.

## MAC assembly on lipid monolayers

For the lipid monolayer assays, the same lipid mixtures and molar ratios were used as for the supported bilayers with 1 mol% of DSPE-PEG(2000)-Biotin. The lipids were mixed, dried under nitrogen flux for 15 min, subsequently put into vacuum for 30 min and dissolved in chloroform (total lipid concentration of 1 mg/ml). The mixtures were further diluted to a final lipid concentration of 0.1 mg/ml and labeled by addition of 0.1 mol% of ATTO655-DOPE (ATTO-TEC GmbH, Siegen, Germany). The total lipid concentration was confirmed by gravimetry.

To form a chamber (see *Scheme 1*), chamber spacers were cut from a 5-mm-thick sheet of PTFE by a laser cutter. The spacers were sonicated step-by-step in acetone, chloroform, isopropanol and ethanol (15 min each). Glass cover slips of 15 mm (Gerhard Menzel GmbH, Braunschweig, Germany) were fixed to the spacer by picodent twinsil 22 two component glue (picodent, Wipperfuerth, Germany). The chambers were washed alternately with ethanol and water, air dried and air plasma-cleaned for 10 min in order to make the glass hydrophilic. The surface was then passivated by covering the glass surface with PLL-PEG(2000) (SuSos AG, Dübendorf, Switzerland) 0.5 mg/mL solution in PBS buffer and incubated for minimum half an hour. After through wash with water (5 times 200 µl) and reaction buffer (3 times 200 µl), the chambers were ready to use.

Lipid monolayers were formed by drop-wise deposition of the lipid mixture on the buffer-air interface (for further details, see also [*Chwastek and Schwille, 2013*]). A lipid mixture volume corresponding to a lipid surface density of 70 Å$^2$ / molecule was deposited drop-wise on the surface of the buffer solution.

The samples were imaged using a Yokogawa scan head CSU10-X1 spinning disk system connected to a Nikon Eclipse Ti inverted microscope (Nikon, Japan) with an Andor Ixon Ultra 512 × 512 EMCCD camera and a 3i solid state diode laser stack with 488 nm, 561 nm and 640 nm laser lines (3il33, Denver, Colorado USA). For simultaneous Alexa-488-phalloidin and ATTO655-DOPE excitation, the 488 nm and the 640 nm laser lines and an UPLanSApo 60x/1.20 Water UIS2 objective (Olympus, Japan) were used. The time interval between the recorded images was 20 s.

After confirming the formation of a phase-separated lipid monolayer by imaging, 100 µl of neutravidin solution (0.01 µg/µl) was added to the sample twice and was incubated for 5 min. Note that all protein solutions or other solutions are applied directly to the liquid subphase by dipping the pipette tip through the monolayer. Next, the subphase was washed five times with buffer (100 µl steps) to remove unbound neutravidin. Subsequently, 20 µl of Alexa-488-phalloidin-labeled actin filaments (2 µM) was added to the subphase and incubated for at least 60 min, since

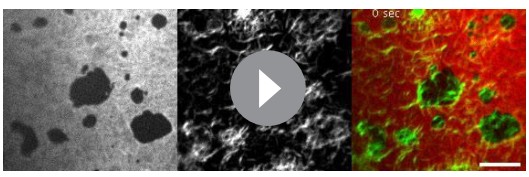

**Video 8.** Shape changes and fusion events during actomyosin contraction of $L_c$ domains in a MAC combined with a phase-separated lipid monolayer. Myofilaments in the presence of ATP led to the contraction of the actin layers and to shape changes and fusion events of the $L_c$ domains. The phase-separated lipid monolayer (3:3:1 DOPC:PSM:Cholesterol) containing the ATTO655-DOPE-labeled $L_e$ phase (red) is shown in the left channel. The middle channel shows Alexa-488-phalloidin-labeled actin filaments that bind to the $L_e$ phase. The right channel shows the merge of both channels. Confocal Spinning Disk image sequence was acquired at 20 s. time intervals and contains 64 frames. The video is displayed at 15 frames per second (fps). Total time: 21 min. Scale bar, 10 µm. Corresponds to *Figure 4C*. (Compressed JPG avi; 1.7 MB).

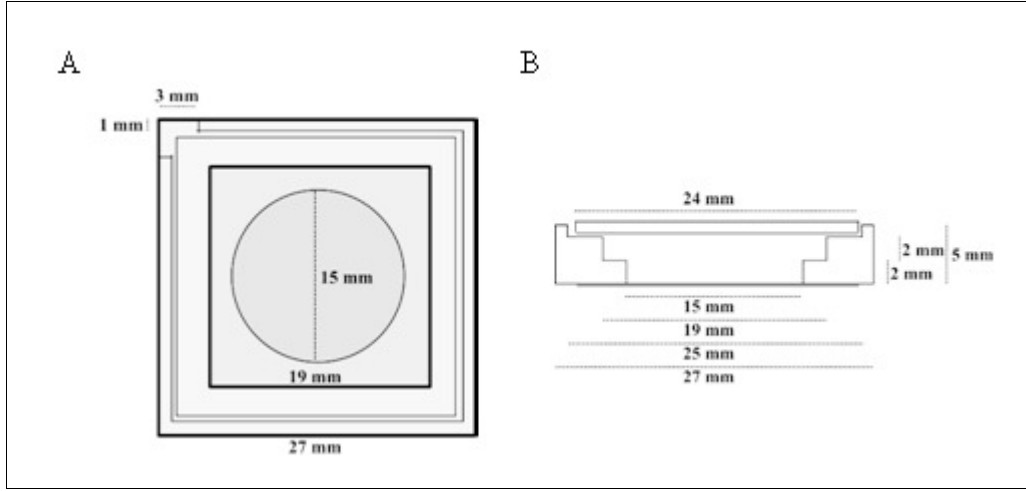

**Scheme 1.** Top (A) and lateral (B) schematic view of the PTFE chamber (adapted from [*Chwastek and Schwille, 2013*]).

binding of actin filaments to the interface was assumed to occur relatively slowly. When the binding of the actin filaments was confirmed by imaging, the monolayer was thoroughly washed (7 to 10 steps of 100 µl) with reaction buffer containing 1 µM ATP and 100 µl of myofilaments (0.3 µM) containing 1 µM ATP (enzymatically regenerated see above) was added to the subphase twice. The sample was sealed by a glass cover slide with grease to avoid subphase evaporation, allowing for long sample observation. The lipid monolayer MAC system started to contract after a few minutes of incubation, resulting in the formation of actomyosin clusters and deformation of the lipid domains.

# Acknowledgements

We are grateful for the financial support by the Daimler und Benz foundation (Project Grant PSBioc8216), the Gottfried Wilhelm Leibniz-Program of the DFG (SCHW716/8-1), the support of the Graduate School of Quantitative Biosciences Munich and the MaxSynBio consortium, which is jointly funded by the Federal Ministry of Education and Research of Germany and the Max Planck Society.

# Additional information

## Funding

| Funder | Grant reference number | Author |
| --- | --- | --- |
| Daimler und Benz Stiftung | 32-09/11 | Sven Kenjiro Vogel |
| Max-Planck-Gesellschaft | | Sven Kenjiro Vogel<br>Ferdinand Greiss<br>Alena Khmelinskaia<br>Petra Schwille |
| Bundesministerium für Bildung und Forschung | | Sven Kenjiro Vogel<br>Petra Schwille |
| Deutsche Forschungsge-meinschaft | SCHW716/8-1 | Sven Kenjiro Vogel<br>Petra Schwille |

The funders had no role in study design, data collection and interpretation, or the decision to submit the work for publication.

## Author contributions
SKV, Conceptualization, Formal analysis, Supervision, Investigation, Writing—original draft; FG, Conceptualization, Formal analysis, Investigation, Writing—original draft; AK, Investigation, Writing—review and editing; PS, Funding acquisition, Writing—review and editing

## Author ORCIDs
Sven Kenjiro Vogel, http://orcid.org/0000-0003-2540-5947
Petra Schwille, http://orcid.org/0000-0002-6106-4847

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
