## [Decision Letter]

Thank you for submitting your article "Control of Lipid Domain Organization by a Biomimetic Contractile Actomyosin Cortex" for consideration by *eLife*. Your article has been reviewed by two peer reviewers, Aurelien Roux (Reviewer #1) and Matthieu Piel (Reviewer #2), and the evaluation has been overseen by Mohan Balasubramanian as the Reviewing Editor and Randy Schekman as the Senior Editor.

As you will see, the expert referees have discussed the reviews with one another and the Reviewing Editor has drafted a series of comments that we request you respond to before we reach a binding decision. The Editor and reviewers request that you address the issue of novelty and the additional experiments they deem essential to elevate this work to the level of an Advance. Please indicate exactly how you propose to proceed and provide an estimate of the time it will take to complete this work. The Editor and the reviewers will then confer and make a recommendation.

The referees appreciated your study on the control of lipid domain organization by the actomyosin cortex. In particular, the referees found that the conclusion that an actomyosin network bound to specific lipids can reorganize lipid domains not bound to such an actomyosin network interesting. They also appreciated the physical model presented in the manuscript. However, it was felt that there were three major issues with the results and the model proposed in this manuscript, which required significant revisions (experiments, rewriting, proper citation of the literature).

Please find below consolidated comments from the referees, following discussion.

Essential revisions:

1) Many of the results have been already published by other groups. I am not against publication of results similar or identical to the ones already published (by other groups of course) as I think documented reproducibility is essential, and usually rare in our publication system. But there must be at least either some new, unreported facts, or a different analysis proposing a different mechanism. The selectivity of the phase, and the nucleation of lipids domains by actin filaments have already been reported, with very similar in vitro conditions by Honigmann et al. *eLife* 2014. The remodeling of lipid domains by active actomyosin network, in particular domain splitting and size selection, have been reported by Koster et al. PNAS 2016. In this manuscript, the monolayer experiments and the model proposed in the Discussion could be the new parts when compared to previous publications, but I am not (yet) convinced that the monolayer system can be easily compared to the supported bilayer (see point 3), nor that the model is supported by experimental data (see point 2). I am thus balanced on this point, as I think that a further effort form the author's on showing their point could revert my opinion.

2) Line tension and the model. I am a bit confused by the proposal of the author. If I understood well, the authors propose that the higher the line tension is, the more energy (pulling force) is necessary to induce the phase transition, which explains why the contour length is changing more for higher temperatures (lower line tension). However, I don't see which experiments or analysis in this manuscript allow to discriminate the author's hypothesis from the one of Koster et al. PNAS 2016 in which the active forces of the actomyosin allow domain splitting and domain fusion.

In fact, the authors report here size selection of domains, which was also reported by Koster et al., which proposed that small domains would preferentially fuse, and large domains would be preferentially broken, leading to a quite narrow range of sizes. Moreover, this manuscript also reports domain splitting and fusion, consistently with Koster et al. hypothesis. Finally, in Koster et al. study, increased line tension leads to lower remodeling, as it counteracts both domain splitting and domain fusion, which could explain the lower contour length increase when line tension is higher, as observed by the authors. But I don't really understand how size selection would work with the author's model that active pulling triggers a phase transition of lipids.

In short, all observations made by Koster et al. are consistent with pure remodeling of the domains, without phase transition, and all the observations are reproduced by the authors. However, I cannot see an experimental result that can discriminate between the two hypotheses, and/or prove their model right.

In Koster et al., a very convincing point is that the total area of the two phases do not change upon actomyosin addition, supporting the hypothesis of Koster et al. that actomyosin only remodel the lipid domains, and does not change the proportion of each lipid phase. This analysis lacks here, and would indeed be essential for proving the point of the authors: if actomyosin was inducing a lipid phase transition, one would expect the total area of the 2 phases (or the surface ratio of the two phases, L_o_ and L_d_) to change upon actomyosin addition. I guess this analysis should be quite straight forward to do in the authors' experimental and should be conducted.

3) The monolayer assay. First, as said by the authors in the text, but not detailed (only the Materials and methods) the conditions used here are fairly different. Instead of a mix DOPC, DPPC and chol as in the bilayer assay, they used C16 shingomyelin, DOPC and chol in very different proportions than DOPC/DPPC/chol. As the chemical interactions between cholesterol and sphingomyelin that participate in phase segregation are very different from the simple, entropy driven, mechanism of DOPC/DPPC separation, I wonder how the two systems can be compared. In fact, the authors see that their lipid-actin linker (DSPE and neutravidin) do not segregate the same than in the supported bilayer assay, supporting my opinion that these two systems cannot be compared easily.

I think it would be essential to perform either the supported bilayer assay with DOPC/Shingo/Chol or the monolayer assay with DOPC/DPPC/Chol. Also in this assay, it would be very interesting to test the authors' model by following the lateral pressure within the monolayer (which can be easily done in these Langmuir films), as the phase transition proposed by the authors should lead to increased pressure. Of course, appropriate controls would also be needed (in particular actin on non-phase separated membranes, as binding of actin to membrane is also expected to act on lipid pressure).

However, measuring the lateral pressure may not reach a clear conclusion on whether actomyosin leads to a phase transition or just lipid domain remodeling. In monolayers at the water/air interface, it is not clear how densely packed the lipids are. Usually in those monolayer assays, it is very difficult to reach the same tight packing than in a bilayer, and a change of total area of each phase can be the result of a phase transition, or compaction/extension of the lipids within a phase without phase transition. Even if evolution of pressure with phase transition is easily understood for system with one lipid, with such complex system the interpretation may reveal itself difficult. And because the total area covered by each phase depends on the lipid packing, the total area analysis proposed above for the supported bilayer may give non-reliable results for the monolayer.

Thus, the monolayer assay, which is probably the newest assay in this manuscript, seems to me having intrinsic flaws for testing the authors' model. The model itself, even if very original and interesting, lacks experimental support at this point, and all other findings reported in this manuscript are consistent with previously published results and models.

---

## [Author Response]

*Essential revisions:*

*1) Many of the results have been already published by other groups. I am not against publication of results similar or identical to the ones already published (by other groups of course) as I think documented reproducibility is essential, and usually rare in our publication system. But there must be at least either some new, unreported facts, or a different analysis proposing a different mechanism. The selectivity of the phase, and the nucleation of lipids domains by actin filaments have already been reported, with very similar in vitro conditions by Honigmann et al. eLife 2014. The remodeling of lipid domains by active actomyosin network, in particular domain splitting and size selection, have been reported by Koster et al. PNAS 2016. In this manuscript, the monolayer experiments and the model proposed in the Discussion could be the new parts when compared to previous publications, but I am not (yet) convinced that the monolayer system can be easily compared to the supported bilayer (see point 3), nor that the model is supported by experimental data (see point 2). I am thus balanced on this point, as I think that a further effort form the author's on showing their point could revert my opinion.*

We thank the reviewer(s) for this argument, which gives us the opportunity to better emphasize the particular aspects of novelty of our study with respect to the related works by Koster et al. PNAS 2016 and Honigmann et al. *eLife* 2014.

Most importantly, our study provides, for the very first time, a direct visualization of lipid domain fission, fusion and movements during actomyosin contractions. Although this has been proposed to occur previously and has been indirectly inferred from the experiments by Koster et al., there has not been any data provided showing the process in real time under controlled temperature conditions. Second, our study presents a technical advance in combining the minimal actin cortices (MACs) with a lipid monolayer system.

The article of Koster et al. deals with a different question and touches upon the issue of phase-separated membranes with a single paragraph and figure only. Neither the main manuscript (Figure 5 page 7) nor the Supporting Information (Figure S5 (page 8) and movie S5) presents a direct visualization of fission, fusion and directed movement of lipid domains. Koster et al. discussed (page 6, lower right) a decrease in the net area of L_o_ domains in the presence of a remodeling actomyosin network (Figure 5 page 7). It is here that they very briefly hypothesized net area domain reduction (page 7 upper left) as follows: “This is likely to be due to a loss of small domains […] by the local remodeling of the actomyosin network in combination with a cessation in growth of larger domains”. In the main body of the article, events such as lipid domain fission, fusion and movements (which we consider our most relevant experimental results) were neither mentioned nor directly visualized.

Honigmann et al. *eLife* 2014, on the other hand, did not report on any actomyosin induced active phase remodeling process, as it deals with static actin filaments. The only changes induced here were temperature changes along the phase diagram of the membrane.

We thank the reviewers for appreciating the combination of the MAC with the monolayer system as a new technical advance. Yet, it was not our intention to employ the lipid monolayer for a quantitative comparison with the supported bilayer system, as this would be hardly possible due to its different nature. We introduced it to minimize friction effects induced by the solid support that could counterbalance the forces induced by the active MAC. Its successful introduction furthermore emphasizes the wide applicability of the MAC system in different lipid assays. We could make use of the difference in actin filament binding to the liquid extended (disordered) L_e_ phase. In low and medium MACs on monolayers, actin filaments close to the phase boundaries align to the circular shape of the liquid condensed (ordered) L_c_ domains because line tension dominates over the actin filament wetting at the phase boundary. Although, again, a quantitative comparison between the supported lipid bilayers and the lipid monolayer was not our intention, we emphasize that we were able to directly visualize lipid domain deformations upon actomyosin contraction in both systems.

*2) Line tension and the model. I am a bit confused by the proposal of the author. If I understood well, the authors propose that the higher the line tension is, the more energy (pulling force) is necessary to induce the phase transition, which explains why the contour length is changing more for higher temperatures (lower line tension).*

To make the model more clear we changed the scheme of the model accordingly (now Figure 4).

We changed the Discussion part accordingly:

“In a model we would first consider a biotinylated lipid that is dragged by an actomyosin filament (Figure 4). […] Further experiments should elaborate on the physical model since actomyosin contraction is clearly a non-equilibrium process and line tension itself was reported to depend on the applied lateral tension (Akimov et al., 2007).”

*However, I don't see which experiments or analysis in this manuscript allow to discriminate the author's hypothesis from the one of Koster et al. PNAS 2016 in which the active forces of the actomyosin allow domain splitting and domain fusion.*

*In fact, the authors report here size selection of domains, which was also reported by Koster et al., which proposed that small domains would preferentially fuse, and large domains would be preferentially broken, leading to a quite narrow range of sizes. Moreover, this manuscript also reports domain splitting and fusion, consistently with Koster et al. hypothesis.*

In contrast to Koster et al. we found for two different lipid mixtures that the area of the L_o_ phase remained constant over the course of actomyosin contraction. Two graphs (Figure 3, left panels) have been added to Figure 3 (see more detailed answers below).

*Finally, in Koster et al. study, increased line tension leads to lower remodeling, as it counteracts both domain splitting and domain fusion, which could explain the lower contour length increase when line tension is higher, as observed by the authors. But I don't really understand how size selection would work with the author's model that active pulling triggers a phase transition of lipids.*

We changed the Discussion part accordingly (see answers above).

*In short, all observations made by Koster et al. are consistent with pure remodeling of the domains, without phase transition, and all the observations are reproduced by the authors. However, I cannot see an experimental result that can discriminate between the two hypotheses, and/or prove their model right.*

*In Koster et al., a very convincing point is that the total area of the two phases do not change upon actomyosin addition, supporting the hypothesis of Koster et al. that actomyosin only remodel the lipid domains, and does not change the proportion of each lipid phase. This analysis lacks here, and would indeed be essential for proving the point of the authors: if actomyosin was inducing a lipid phase transition, one would expect the total area of the 2 phases (or the surface ratio of the two phases, L_o_ and L_d_) to change upon actomyosin addition. I guess this analysis should be quite straight forward to do in the authors' experimental and should be conducted.*

We performed the analysis of the changes in net domain area for both used lipid mixtures (1:2:1 DOPC:DPPC:Cholesterol and 3:3:1 DOPC:PSM:Cholesterol). We found for both mixtures that the area of the L_o_ phase remained constant over the course of actomyosin contraction. Two graphs (Figure 3, left panels) have been added to Figure 3. In contrast to the work of Koster et al., we performed the analysis on two different lipid mixtures and did not observe a decrease of the total L_o_ phase domain area in neither of the lipid mixtures.

We revised the text in the Results part accordingly:

“For both mixtures the area of the L_o_ phase remained constant over the course of actomyosin contraction (Figure 3, left panels). […] In contrast to their work we did not observe a decrease of the total L_o_ phase domain area in neither of the lipid mixtures (Figure 3, left panels).”

*3) The monolayer assay. First, as said by the authors in the text, but not detailed (only the Materials and methods) the conditions used here are fairly different. Instead of a mix DOPC, DPPC and chol as in the bilayer assay, they used C16 shingomyelin, DOPC and chol in very different proportions than DOPC/DPPC/chol. As the chemical interactions between cholesterol and sphingomyelin that participate in phase segregation are very different from the simple, entropy driven, mechanism of DOPC/DPPC separation, I wonder how the two systems can be compared. In fact, the authors see that their lipid-actin linker (DSPE and neutravidin) do not segregate the same than in the supported bilayer assay, supporting my opinion that these two systems cannot be compared easily.*

*I think it would be essential to perform either the supported bilayer assay with DOPC/Shingo/Chol or the monolayer assay with DOPC/DPPC/Chol.*

We performed the suggested experiments with the two lipid mixtures for the lipid bilayer and lipid monolayer systems. The results of the new lipid mixture for the supported lipid bilayer (SLB) system (3:3:1 DOPC:PSM:Cholesterol) were added to Figure 3 (Figure 3; Video 5–Video 7). For both lipid mixtures we directly visualized various shape changes of lipid domains upon actomyosin contractions indicating the general validity of our experimental findings. For both lipid mixtures the net area of the L_o_ phase remained constant (see above; Figure 3, left panels).

We revised the text in the Results part accordingly:

“For studying the effects of myofilament induced actin rearrangements we prepared two lipid mixtures for lipid mono- and bilayer experiments, namely DOPC, DPPC and Cholesterol in a 1:2:1 molar ratio and DOPC, PSM and Cholesterol in a 3:3:1 molar ratio containing 0.1 or 1 mol% DSPE-PEG(2000)-Biotin DSPE (see Materials and methods[…] In summary, we showed active lateral reorganization of lipid domains for two different lipid mixtures, indicating the general validity of our experimental observations.”

In addition, we also performed new experiments with the lipid monolayer system using the 1:2:1 DOPC:DPPC:Cholesterol lipid mixture herewith using the same lipid mixtures as for the SLB system. Also in the additional lipid mixture we found that contrary to SLB system, actin filaments preferentially bind to the liquid extended (disordered) (L_e_) phase where also the neutravidin anchor protein mainly partitioned to (see Figure 5). We observed that in both systems actin filaments close to the phase boundaries align to the circular shape of the L_c_ domains indicating that the line tension energy dominates over the actin filament wetting energy at the phase boundary (see Author response image 1AD and Figure 4 13 in the revised manuscript). In both lipid mixtures the actomyosin contractions led to shape changes and rearrangement of the L_c_ domains including fusion and stretching events (Figure 4; Video 8). We implemented a new figure (Figure 4) to the revised manuscript emphasizing the domain shape changes and fusing events in the lipid monolayer system during actomyosin contraction. Since the additional lipid mixture did not add new qualitative information to the monolayer system we decided not to include the new results in the revised manuscript, yet the data is shown in Figure 5.

We revised the text in the Results part accordingly:

“Contrary to the situation in the SLB system, actin filaments preferentially bind to the liquid extended (disordered) (L_e_) phase where also the neutravidin anchor protein mainly partitioned to (Figure 4). We observed this behavior for both lipid mixtures that were also used for the SLBs (see Materials and methods) […] The addition of myosin filaments in the presence of ATP led to the contraction of the actin layer deforming and rearranging the L_c_ domains in both lipid mixtures similar to what we observed in the SLB systems (Figure 4; Video 8). The shape changes included fusion and stretching of the L_c_ domains (Figure 4; Video 8).”

Author response image 1.Confocal Spinning Disk images of the additional lipid monolayer mixture (1:2:1 DOPC:DPPC:Cholesterol).Actin filaments close to the phase boundaries align to the circular shape of the liquid condensed (ordered) L_c_ domains because the line tension energy dominates over the actin filament wetting energy at the phase boundary. The same behavior was observed for the 3:3:1 DOPC:PSM:Cholesterol lipid mixture (Figure 4). Scale bars, 10 µm.**DOI:**
http://dx.doi.org/10.7554/eLife.24350.015

*Also in this assay, it would be very interesting to test the authors' model by following the lateral pressure within the monolayer (which can be easily done in these Langmuir films), as the phase transition proposed by the authors should lead to increased pressure. Of course, appropriate controls would also be needed (in particular actin on non-phase separated membranes, as binding of actin to membrane is also expected to act on lipid pressure).*

*However, measuring the lateral pressure may not reach a clear conclusion on whether actomyosin leads to a phase transition or just lipid domain remodeling. In monolayers at the water/air interface, it is not clear how densely packed the lipids are. Usually in those monolayer assays, it is very difficult to reach the same tight packing than in a bilayer, and a change of total area of each phase can be the result of a phase transition, or compaction/extension of the lipids within a phase without phase transition. Even if evolution of pressure with phase transition is easily understood for system with one lipid, with such complex system the interpretation may reveal itself difficult. And because the total area covered by each phase depends on the lipid packing, the total area analysis proposed above for the supported bilayer may give non-reliable results for the monolayer.*

*Thus, the monolayer assay, which is probably the newest assay in this manuscript, seems to me having intrinsic flaws for testing the authors' model. The model itself, even if very original and interesting, lacks experimental support at this point, and all other findings reported in this manuscript are consistent with previously published results and models.*

We agree that a thorough study on the influence of lipid packing on the domain modulation by actin reorganization would be highly interesting. However, as correctly pointed out, this may be difficult to interpret and is beyond the scope of this article. The monolayer assay was introduced for qualitative comparison without the influence of friction and not as a better test bed for the model.